Journal of Data-centric Machine Learning Research (2024)          Submitted 2/24; Revised 7/24; Published 8/24

# VALUED - Vision and Logical Understanding Evaluation Dataset

**Soumadeep Saha**                                    SOUMADEEP.SAHA_R@ISICAL.AC.IN
*Indian Statistical Institute*
*Kolkata, India*

**Saptarshi Saha**                                    SAPTARSHI.SAHA_R@ISICAL.AC.IN
*Indian Statistical Institute*
*Kolkata, India*

**Utpal Garain**                                    UTPAL@ISICAL.AC.IN
*Indian Statistical Institute*
*Kolkata, India*

**Reviewed on OpenReview:** *https: // openreview. net/ forum? id= nS9oxKyy9u*

**Editor:** Sergio Escalera

## Abstract

Starting with early successes in computer vision tasks, deep learning based techniques have since overtaken state of the art approaches in a multitude of domains. However, it has been demonstrated time and again that these techniques fail to capture semantic context and logical constraints, instead often relying on spurious correlations to arrive at the answer. Since application of deep learning techniques to critical scenarios are dependent on adherence to domain specific constraints, several attempts have been made to address this issue. One limitation holding back a thorough exploration of this area, is a lack of suitable datasets which feature a rich set of rules. In order to address this, we present the VALUE (Vision And Logical Understanding Evaluation) Dataset, consisting of 200,000+ annotated images and an associated rule set, based on the popular board game - chess. The curated rule set considerably constrains the set of allowable predictions, and are designed to probe key semantic abilities like localization and enumeration. Alongside standard metrics, additional metrics to measure performance with regards to logical consistency is presented. We analyze several popular and state of the art vision models on this task, and show that, although their performance on standard metrics are laudable, they produce a plethora of incoherent results, indicating that this dataset presents a significant challenge for future works.

**Keywords:** logical constraints, domain knowledge, deep learning, computer vision

## 1 Introduction

Incorporating domain-knowledge has been highlighted as one of the "3 Grand Challenges in developing AI systems" by a recent report on AI for Science (Stevens et al., 2020) and has been studied with zeal across several domains. Although domain-knowledge can refer to various kinds of information (relevant features, known rules, probability distributions, etc), following the survey by Dash et al. (2022), we restrict our discussion to *problem specific prior*

*knowledge, represented as logical or numeric constraints.* These constraints take several forms depending on the domain in question, like semantic faithfulness (Chaturvedi et al., 2022) or reasoning tasks (Srivastava et al., 2023) in natural language processing (NLP), diagnostic constraints (Saha et al., 2023b) in clinical settings, graph based constraints in functional genomics (Yu et al., 2018; Kulmanov et al., 2017) and general propositional logic rules in a multitude of settings (Kowsari et al., 2020; Agrawal et al., 2013; Mayne and Perry, 2009). For example, consider rules like - *"a scene can contain at most 10 objects"* (numerical constraint) or *"a scene cannot contain both objects of type A and B"* (logical constraint) which might serve as "prior knowledge" for a computer vision task. Imbuing deep learning systems with the ability to adhere to these constraints would greatly improve their applicability to critical areas like robotics, healthcare, law, or material science. Additionally, development of such techniques should in principle reduce reliance on massive datasets, which are especially difficult to annotate in these contexts (Yang et al., 2019).

To illustrate this problem, consider the example of a vision system employed to read credit card numbers in order to process payments. Although, digit recognition is a well understood task, even 1% misclassification in a standard 16 digit card number can result in 14.9% of transactions failing. This is a common issue, and logical constraints in the form of checksums(ISO/IEC 7812-1:2017) are employed to limit inaccuracies. A model armed with this knowledge could avoid making erroneous predictions, or potentially even infer ambiguous digit. Even commonplace logical inferences like - *(A is inside B) and (B is on top of C)* $\Rightarrow$ *A is on top of C*, which we take for granted in everyday life, potentially might not be captured in deep learning based computer vision systems trained on large datasets (Johnson et al., 2017), and might present challenges for downstream applications. Today, with the proliferation of applications for deep learning based vision systems, there is a pressing need to evaluate logical understanding and further investigate possible avenues of inculcating them into trained models.

Since training deep neural networks is reliant on gradient based methods, and logical/numeric constraints are in general discrete, we run into our very first hurdle. It is not a priori clear how standard techniques should be adapted to abide by these constraints, and due to its significant practical importance, this field has been the subject of extensive research (Evans and Grefenstette, 2018; Hu et al., 2016a; Riegel et al., 2020) (we refer the reader to surveys by Besold et al. (2017) or von Rueden et al. (2023) for a comprehensive overview). In spite of this, a standard framework for incorporating logical constraints into neural networks eludes us (Dash et al., 2022).

One hindrance often encountered across the literature, is in regards to availability of large volumes of high quality annotated data with associated rules (Xie et al., 2021; Muralidhar et al., 2018; Yang et al., 2019). To study the acuity of vision systems in this context, we need to design a task, ideally demonstrating the following characteristics:

1. Poses a challenge for state of the art vision foundation models.

2. Contains non-trivial first-order logic rules that can be inferred from data.

3. Presents semantic constraints on localization to test visual reasoning and numerical constraints to test arithmetic reasoning.

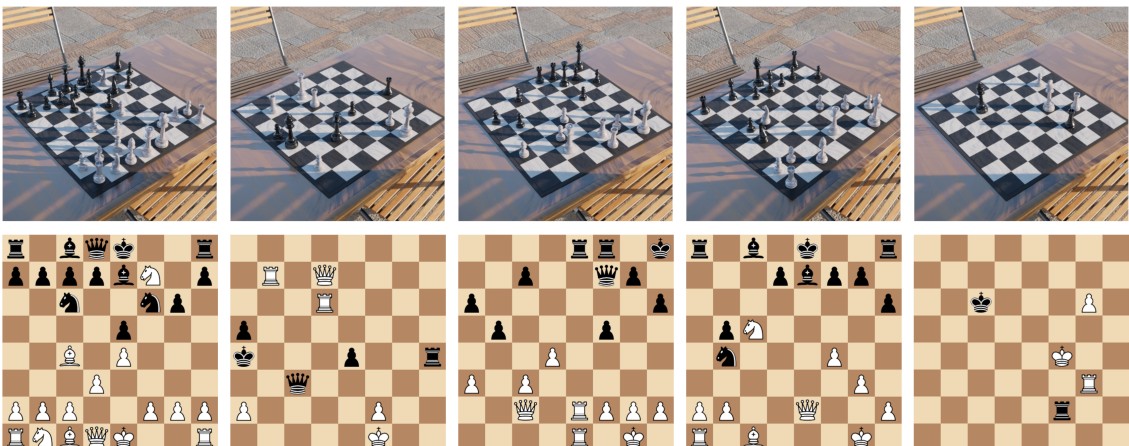

Figure 1: Example images from the VALUE Dataset (*top row*) and corresponding expected outputs (*bottom row*).

With these features in mind we land on the problem of chess board state recognition, i.e. given an image of a chess game in progress, reconstruct the state of the board. Note that our primary interest is not on the problem of accurately identifying chess game states, as in Masouris and van Gemert (2023) or Wölflein and Arandjelović (2021), but rather to analyze the shortcomings of state of the art vision systems in logical understanding and the efficacy of mitigation techniques.

We present the **VALUE Dataset - a collection of 200,000+ annotated images of chess games in progress** (see Figure 1 for examples [1]), with a **curated set of logical constraints**, and constraint adherence metrics designed to analyse performance of **domain-knowledge incorporation techniques**. The rule set, containing several non-trivial first-order logic rules that are necessarily true for a certain prediction to be a valid chessboard state, also enables analysis of models with regards to **understanding of key semantic information** like location, count, etc. This dataset, with the addition of relevant metrics like F1-score and exact match (EM%) is intended to serve as a standard benchmark task, to analyze various algorithms and strategies developed to address logical coherence of deep learning systems. Since data on several million real world chess games is readily available (Org.), and necessary materials to generate images is provided herewith, more data can be easily generated if necessary.

This dataset will aid in the exploration of several pressing questions like, (1) given enough data can deep learning systems trained with standard approaches learn to abide by underlying logic rules (2) how does quality/quantity of incorporated rules influence learning, and (3) how well does a particular technique perform in enforcing logical constraints. Since the issue of logical incoherence is a hindrance to adoption of deep learning techniques to several key real world applications like robotics, automated diagnostics, etc., we present

---

1. For more examples and to download the dataset visit https://espressovi.github.io/VALUED

this dataset to the community with the hope that it promotes further research into the development of techniques to solve this problem.

## 2 Background

Domain-knowledge in the form of logical or numerical constraints is a common feature across several problems. Although these constraints are sometimes continuous/differentiable (Karpatne et al., 2017; Muralidhar et al., 2018), the discrete cases conflict with the reliance on gradient based optimization methods used in deep learning, thus presenting a more interesting challenge. The methods explored to solve this problem usually involves transforming the input data (França et al., 2014), the loss function (Xu et al., 2018), the model itself (Du et al., 2020) or some combination of these (Hu et al., 2016b). However, more work is needed to find a standard framework for incorporating domain knowledge constraints into deep learning based systems (Dash et al., 2022).

One complaint frequently resonating in the literature throughout this area is the dearth of datasets (Yang et al., 2019; Du et al., 2020; Xie et al., 2021). Specifically, annotated datasets featuring domain specific constraints are required to enable analysis and development of techniques. Researchers rely on toy datasets (Serafini and d'Avila Garcez, 2016; Riegel et al., 2020) or datasets containing only a few thousand examples (Melacci et al., 2022).

Often incorporation of domain-knowledge is sold as a cure to this problem, thus results are reported on extremely sparse datasets(Yang et al., 2019; Muralidhar et al., 2018). However, if demonstrations of techniques are limited to sparse datasets, the significance of improvements owing to intervention techniques remains unclear. Domain knowledge in principle, should be able to alleviate performance bottlenecks in low resource problems, but it is by no means a settled question, and the effect of dataset size on the ability to learn domain knowledge needs further exploration.

Recently, some progress has been made in this regard in the NLP domain with the introduction of BIG-bench (Srivastava et al., 2023) (Beyond the Imitation Game benchmark) which introduces a large set of tasks that are constrained by logic, however, owing to the vast complexity of natural language, a comprehensive set of logical constraints covering a significant fraction of the domain is hard to come by. Therefore, it does not address the need for a large dataset with a comprehensive set of rules constraining a large fraction of instances. There have been some efforts in assessing the visual reasoning ability of deep learning based systems (Johnson et al., 2017; Andreas et al., 2016; Zhang et al., 2019). However these datasets asses the models ability for abstract visual reasoning, and to the best of our knowledge there are no high quality datasets analyzing deductive reasoning ability based on a rich set of known domain specific axioms.

Another context where this problem has been widely studied is multi-label classification with logical constraints (Huang et al., 2023; Melacci et al., 2022; Kulmanov and Hoehndorf, 2022). In these problems, a knowledge graph imposes entailment constraints that valid predictions must adhere to, but generally the set of admissible rules used are rather limited in scope. For example, in the Blurb-genre dataset (Aly et al., 2019) or the Uniprot protein function dataset (Dimmer et al., 2011) or Ruepp et al. (2004); Suzgun et al. (2022); Saha et al. (2023a), only rules of the form $\forall x, A(x) \Rightarrow B(x)$ is featured. Richer rule sets that

include constraints reliant on localization, counting, etc in addition to simpler first-order logic rules need to be explored. Some large-scale datasets (Alday et al., 2020; Dimmer et al., 2011) feature a long-tailed class distribution and extremely sparse classes, befogging analysis of domain-knowledge incorporation techniques. To promote and foster further inquiry in this area, large-scale datasets featuring a diverse set of domain specific rules, and unfettered by unrelated confounding issues like sparsity, class imbalance, etc. are needed.

There have been previous attempts at creating datasets of images of chess games by Wölflein and Arandjelović (2021) and Masouris and van Gemert (2023), however their focus was on creating accurate systems for game state recreation, and not analysis of logical understanding, and thus do not comprise of a rule set or associated metrics. Although these datasets feature an order of magnitude fewer images ($\sim$ 5K, 10K respectively) these datasets can be subsumed into ours to increase diversity and further aid in probing logical understanding.

### 2.1 Brief primer on chess

Chess is a two-player board game dating back centuries, played on an 8×8 grid with each player controlling 16 pieces of 6 distinct types. Each distinct piece has constraints on their allowed movements (legal moves) (FIDE) in the grid, and players take turns moving one piece at a time with the objective of putting the opponent in a position where the capture of their king is unavoidable (checkmate).

The two players' pieces are distinguished by their color (black or white), and a single character acronym is assigned to each piece, which are `k,q,r,b,n,p` for king, queen, rook, bishop, knight and pawn respectively (lower case characters refers to black pieces and upper case characters refers to white pieces). Each file (column) of the chessboard (grid) is indexed with a letter in {A, ..., H} starting from the left, and each rank (row) is indexed by a number in $\{1, \ldots, 8\}$ starting from white's side (bottom leftmost dark square for each board, e.g. in Figure 1, is indexed A1). The configuration of the entire board (board state) is often represented in a shorthand notation called Forsyth-Edwards Notation (FEN), which lists the pieces in each rank in sequence separated by a slash ("/"), with blank space counts interspersed in between. For example, in Figure 1 the FEN for the leftmost and rightmost boards are "`r1bqk2r/ppppbN1p/2n2np1/4p3/2B1P3/3P4/PPP2PPP/RNBQK2R`" [2] and "`8/8/2k3P1/8/5K2/6R1/5r2/8`" respectively and the game starts from the state "`rnbqkbnr/pppppppp/8/8/8/8/PPPPPPPP/RNBQKBNR`". We also use |L| to denote the number of pieces of type L on a board (e.g. $|k| = |K| = 1$).

## 3 Task description

If this were a standard classification task, we would seek to learn a function $f_\theta : \mathcal{D} \rightarrow [0, 1)^{(\# \text{ pieces } + 1) \times 8 \times 8}$ (prediction for piece type or empty for all 64 positions) that models the probability of each piece at every board position given an image of the board state. However, when in use, this function must be paired with an inference algorithm with discrete outputs to recreate the board state, which can itself introduce logical inconsistencies. Thus,

---

2. `r1bqk2r/`... represents a row with a black rook, followed by an empty square, followed by a black bishop, queen and king, followed by two empty squares and a black rook arranged from left to right. This is similarly repeated for all ranks (rows)

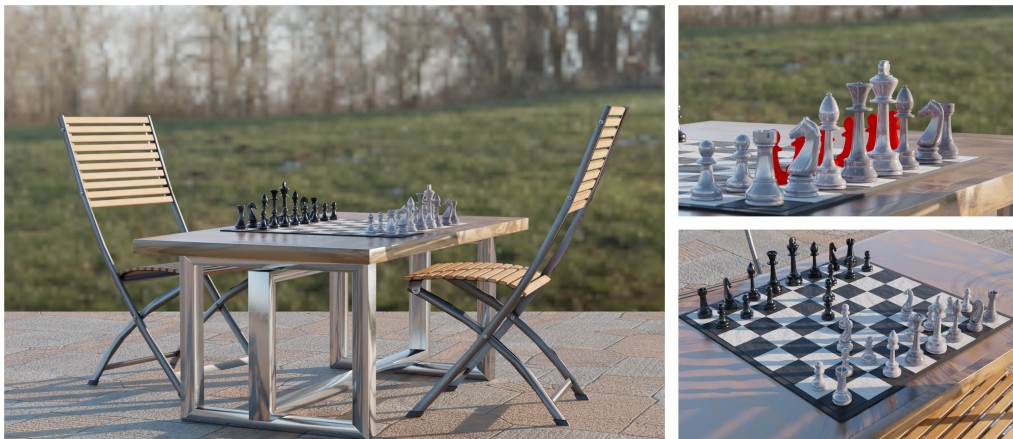

Figure 2: The base 3D scene (*left*) and dataset examples demonstrating occlusion (*marked in red*), and object density (*right*).

in order to analyse performance with regards to logical coherence, the predictive model with soft outputs and the inference algorithm is treated together as a black-box, and performance is evaluated with respect to the complete board state prediction. Keeping this in mind, we define a classifier as follows:

$$F_\theta : \mathcal{D} \rightarrow \mathrm{P}^{8\times 8}$$
$$F_\theta(x) \mapsto \text{board state of } x \tag{1}$$

where $\mathcal{D} \subset \mathbb{R}^{512\times 512\times 3}$ is the space of input images, P = {x, p, P, n, N, b, B, r, R, q, Q, k, K} is the set of all pieces (x represents the empty grid location), and $F_\theta$ represents the model we seek, parametrized by $\theta$. The images pose several difficulties from a computer vision perspective like camera position variability, occlusion (often severe), and dense clusters of similar looking small objects (see Figure 2).

Note that this problem can be rephrased as a semantic segmentation problem or a captioning task or potentially an entirely new paradigm of machine learning, which we don't analyze. This is because, in the context in which we are analyzing the problem, one must imagine that our deep learning system is a part of a larger pipeline that performs some task in the real world, and the outputs from the model are further fed to an automated system. It is in these sorts of applications that demonstration of logical understanding is most pressing. In our sample task, if the model generated annotations were presented to a human with a few errors, they could readily correct those, however, if this model was part of an automated chess-playing robot, that also employs a chess engine to generate moves and software to control linkages, logical errors have a much more pronounced effect. Thus we only analyze this as a classification problem, since any other paradigm would first require conversion to a standardised format identical to one defined in Equation 1, before further processing.

### 3.1 The rule set

An arrangement of pieces on the chessboard is called **valid**, if there exists a sequence of legal moves (FIDE) from the starting position that results in the arrangement and the set of all valid states is denoted $\mathcal{V}$. Given the state of a chessboard, it is computationally prohibitive to determine if the state is valid, however, some simple *sanity checks* can be utilized to rule out the vast majority of invalid states. We curated such a set of computationally cheap ($\mathcal{O}(nd^2)$ for $n$ chessboards of side $d$ i.e. $\mathcal{O}(n)$) first-order logic rules that hold true for all valid chessboard states, to measure domain coherence. The proposed rule set has the following rules (equivalent rules for white pieces are also included):

$\forall y \in \mathcal{V}$ (valid states) we have,

   i) $|\mathbf{k}| = 1$    (Exactly one king.)

   ii) $\mathbf{k}, \mathbf{K}$ are not on adjacent squares.

   iii) $|\mathbf{p}| + |\mathbf{q}| + |\mathbf{n}| + |\mathbf{b}| + |\mathbf{r}| \leq 15$

           (Total number of pieces including king cannot exceed 16.)

   iv) $|\mathbf{p}| \leq 8$    (Total number of pawns not exceeding 8.)

   v) $\forall \mathbf{p}, 2 \leq rank(\mathbf{p}) \leq 7$    (No pawn on first or last rank.)

   vi) $|\mathbf{p}| = 8 \Rightarrow \left( |\mathbf{q}| \leq 1 \wedge |\mathbf{b}| \leq 2 \wedge |\mathbf{n}| \leq 2 \wedge |\mathbf{r}| \leq 2 \right)$

     (If no pawn is promoted, there cannot be more than two bishops, knights, rooks

          or more than one queen.)

   vii) $|\mathbf{p}| < 8 \Rightarrow$

   $\max(0, |\mathbf{q}| - 1) + \max(0, |\mathbf{b}| - 2) + \max(0, |\mathbf{n}| - 2) + \max(0, |\mathbf{r}| - 2) \leq 8 - |\mathbf{p}|$

     (If pawns might have been promoted, number of excess pieces cannot exceed number

          of missing pawns.)

   viii) $(|\mathbf{p}| = 8) \wedge (|\mathbf{b}| = 2) \Rightarrow$     $\mathbf{b}_1, \mathbf{b}_2$ don't occupy squares of same color

     (If no pawn has been promoted and there are two bishops, they must be on opposite

          color squares.)

$$\tag{2}$$

If a prediction $y$ **satisfies all these rules** we call it **sane**, and we have the **set of sane states** $\mathcal{S}$ following $\mathcal{V} \subsetneq \mathcal{S} \subsetneq \mathrm{P}^{64}$. These rules (Equation 2) are further divided into two categories - **counting** (i, iii, iv, vi, vii), and **localizing** (ii, v, viii), to analyze specific semantic abilities. Counting rules apply constraints on number of objects that can be present in the scene, whereas localizing rules apply constraints on their position in the scene.

    This is not an exhaustive list, and more such rules can be found (e.g. $\mathbf{b}$ can't be trapped in the last rank behind 3 adjacent $\mathbf{p}$, etc.), but these are nonetheless effective at eliminating a large fraction of invalid states. Note that the total number of distinct predictions we would have following standard deep learning approaches (64 independent classification problems) is $|\mathrm{P}^{64}| = 13^{8 \times 8} \sim 10^{72}$, but with the addition of just the constraint on number of pieces, it reduces to $< 10^{55}$.

    If these simple rules can be incorporated into the learning algorithm, in addition to being more applicable to the domain, it could in principle improve performance drastically.

For example, if q was misclassified as k, or b was misclassified as p resulting in $|\mathbf{p}| = 9$, the rule set would identify and seek to disincentivize it.

### 3.2 Evaluation Metrics

We adapt a few standard metrics to evaluate raw performance as well as domain alignment. Since the logical constraints work with the discretized complete board state prediction, and per our definition in Equation 1 "threshold tuning" or related inference methods are considered part of the model, some popular metrics like AUPRC, AUROC or macro-averaged metrics like mAP are not discussed. Given a prediction set $\hat{Y} = \{\hat{y}_i | i \in \{1, \ldots, n\}\}$ and the corresponding ground truth set $Y = \{y_i\}$, $y_i, \hat{y}_i \in P^{64}$, we define exact match (EM) and F1 as follows.

$$EM(Y, \hat{Y}) = \frac{1}{n} \sum_{i=1}^{n} \mathbb{I}([y_i]_k = [\hat{y}_i]_k, \ \forall k, 1 \le k \le 64) \tag{3}$$

$$F1(Y, \hat{Y}) = \frac{2}{n} \sum_{i=1}^{n} f_1(y_i, \hat{y}_i), \text{ where}$$

$$f_1(y_i, \hat{y}_i) = \frac{\sum_{j=1}^{64} \mathbb{I}([y_i]_j = [\hat{y}_i]_j \ne \mathbf{x})}{|y_i| + |\hat{y}_i|} \tag{4}$$

Where $|y| = \sum_j \mathbb{I}([y]_j \ne \mathbf{x})$ denotes the number of non-empty squares in the grid, and $\mathbb{I}$ is the indicator function that takes the value 1 if its argument condition is true, and 0 otherwise.

Additionally, we define two measures of domain coherence – **contradiction %** (C) and **sane F1** (sF1) as follows.

$$C(Y) = \frac{100}{n} \sum_{i=1}^{n} \mathbb{I}(\hat{y}_i \notin \mathcal{S}) \tag{5}$$

$$sF1(Y, \hat{Y}) = \frac{2}{n} \sum_{i=1}^{n} \left( \mathbb{I}(\hat{y}_i \in \mathcal{S}) \cdot f_1(y_i, \hat{y}_i) \right) \tag{6}$$

C reflects the frequency of logical constraint violations i.e. what fraction of predictions are unusable, and sF1 score measures the F1 score after eliminating predictions that are not sane. We also report results on $\mu_C$, the mean number of rule violations per instance in $\hat{Y}$.

$$\mu_C(\hat{Y}) = \frac{1}{n} \sum_{i=1}^{n} \left( \# \text{ of rule violations in } \hat{y}_i \right) \tag{7}$$

When the model is part of an automated system with downstream applications mandating adherence with domain rules, offending predictions must be suppressed with consistency checks against the rule set. The sF1 measure considers this counterfactual scenario, where these rule violating predictions were replaced with a null prediction set. The difference between F1 and sF1 establishes a floor for the room for improvement available to algorithms seeking to incorporate logical constraints.

To summarize, in this task, we seek an $F_\theta$, such that given a set of images $x \in \mathcal{D}$, it can faithfully recreate corresponding ground truth labels $y \in \mathrm{P}^{64}$, while minimizing violations of domain rules, i.e. $sF1\Big(\{F_\theta(x_i)\}, \{y_i\}\Big)$ is maximized.

## 4 Methods

To create the dataset [3], we first procured a large database of chess moves in online multi-player games from lichess (Org.) and converted them to board states. Although this database of board states contains duplicates (e.g. common openings), we do not remove them in order to preserve the distribution of states that are likely to be encountered. A 3D model of the world consisting of a large plane with a table, two chairs, a chess board and pieces in starting position (see Figure 2) was created in the open-source modelling software blender (Community, 2018). The board state database was then used to move the pieces to their designated squares (with some noise). A camera was added to the scene at a fixed distance from the center of the chessboard, constrained to point to the center, with randomized pan and tilt. The camera motions are limited to ensure that the chessboard corner A1 is closest to the camera to avoid ambiguity with respect to rotations of the board. 200,000 such images were rendered at $(512 \times 512 \times 3)$ to form the training/validation set and an additional 19,967 images form the test set. In addition to the labels (in array form and FEN), bounding box information for every piece is provided to aid other techniques (object detection, etc.) but these are not used for evaluation. A concise rule set was created to strike a balance between imposing sufficient constraints and computationally efficient checking. Furthermore, the rule set implementation provides analysis on the type of errors being made by the prediction system.

To establish baseline results we selected a range of popular ImageNet (Deng et al., 2009) pre-trained vision models like ResNets (He et al., 2016), ViT (Dosovitskiy et al., 2021), etc. covering a large range of scales and training techniques. These pre-trained models were fine-tuned for 2 epochs after replacing the final fully connected layer with one of size ($\mathtt{in\_features} \to 8 \times 8 \times \mathtt{class\_number}$) and adding dropouts(10%) in between. The models were implemented in pytorch, and trained with the AdamW optimizer (learning rate of $10^{-4}$) and Cross Entropy loss. The images were normalized, and resized if necessary. We trained the models on a single NVIDIA A6000 48GB GPU. All associated code (database creation, rule checking, etc.), materials (dataset, 3D models, textures, images, etc.), rendering details (camera sensor, rendering settings, etc.) and relevant information has been made available through the github repository.[4]

## 5 Baseline results and discussions

Performance analysis results of various vision models on our dataset is summarized in Table 1. Although these pre-trained models feature a range of training techniques and scales, they are quite adept at recognizing relevant visual features in order to recognize chess pieces as exemplified by their high F1 and EM scores (see Table 1). Notably, the Swin Transformer (Liu et al., 2021) outperforms models which are an order of magnitude larger, perhaps

---

3. Dataset - https://doi.org/10.5281/zenodo.10607059
4. Code repository https://github.com/espressoVi/VALUE-Dataset

because of its ability to capture features at different scales owing to its standout hierarchical architecture. Still, these models leave a lot to be desired from a domain consistency point of view, as seen by the large percentage of predictions that have rule violations. In critical applications, these incoherent predictions aren't viable, and would have to be discarded, which is reflected in the significantly reduced sF1 scores. To further emphasise the point we direct the readers' attention to $\mu_C$ in Table 1, which records the mean number of rule violations per prediction, where, in the best case scenario we encounter a rule violation every 15 predictions and every 2.3 predictions in the worst case. Although we reported results with off the shelf vision systems, we note that the performances of systems designed for the specific chess board state recognition task are also comparable, with $EM$ scores of $\sim15\%$ and $\sim7\%$ for an end to end and piece-wise classification system respectively (Masouris and van Gemert, 2023).

Table 1: Performance of popular vision models on VALUE dataset. ($[\uparrow]$ - higher is better, $[\downarrow]$ lower is better). (1) - Simonyan and Zisserman (2015) (2) - He et al. (2016) (3) - Dosovitskiy et al. (2021) (4) - Liu et al. (2021)

| Model | Param. Num. | Image Size | EM (%)[$\uparrow$] | C (%) [$\downarrow$] | F1 [$\uparrow$] | sF1 [$\uparrow$] | F1-sF1 [$\downarrow$] | $\mu_c$ [$\downarrow$] |
|---|---|---|---|---|---|---|---|---|
| VGG-16 (1) | 134M | $224^2$ | 26.3% | 28.7% | 0.880 | 0.656 | **0.224** | 0.426 |
| ResNet50 (2) | 24M | $512^2$ | 56.3% | 12.8% | 0.959 | 0.849 | **0.110** | 0.172 |
| ResNet101 (2) | 43M | $512^2$ | 60.4% | 11.1% | 0.966 | 0.869 | **0.097** | 0.147 |
| ViT-B/16 (3) | 86M | $224^2$ | 25.9% | 30.8% | 0.875 | 0.635 | **0.240** | 0.432 |
| ViT-L/32 (3) | 307M | $384^2$ | 32.2% | 24.0% | 0.907 | 0.711 | **0.196** | 0.337 |
| SWIN-tiny/4 (4) | 29M | $224^2$ | 80.3% | 5.2% | 0.984 | 0.938 | **0.046** | 0.067 |

Since our rule set features various rules designed to test localization ability, i.e. constraints on where objects can be located in the scene, and counting ability, i.e. constraints on how many objects there can be in the scene, we analyzed model performance in this regard. Although the vast majority of errors across all models arise due to violation of counting constraints (see Figure 3a), note that the likelihood of making counting errors is higher owing to the much larger number of possible predictions where such a constraint is violated. To adjust for this we compute-

$$P^{\mathrm{R}}_{model} = f^{\mathrm{R}}_{model}/f^{\mathrm{R}}_{random} \qquad (8)$$

where $f^{\mathrm{R}}_{model}$ is the frequency of rule R being violated by the model in question, and $f^{\mathrm{R}}_{random}$ is the frequency of rule violation of a random guesser ($P(s) = P(s'), \forall s, s' \in \mathrm{P}^{64}$). This shows (see Figure 3b) that the models are more likely to make errors in regards to localization ($\mu = 0.0052 \pm 0.0025$) than counting ($\mu = 0.0042 \pm 0.0023$).

Further, we probed the models to check for the likelihood of violating each rule under the two categories. To our surprise, rule (v) in Equation 2, was never violated by any of the models, and rules (iii), (vii) are extremely unlikely to be violated (see Figure 3c). On the other hand, locality constraints (ii), (viii), and counting rules (i), (iv), (vi) are extremely

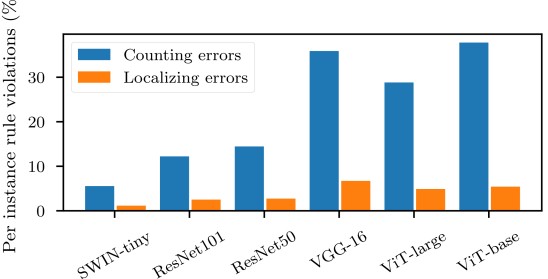

(a) Prevalence of violating counting or localizing rules.

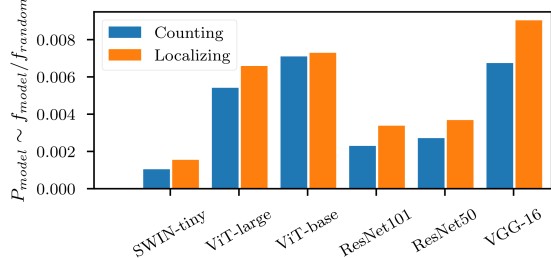

(b) Adjust likelihood of violating counting or localizing rules (Equation 8).

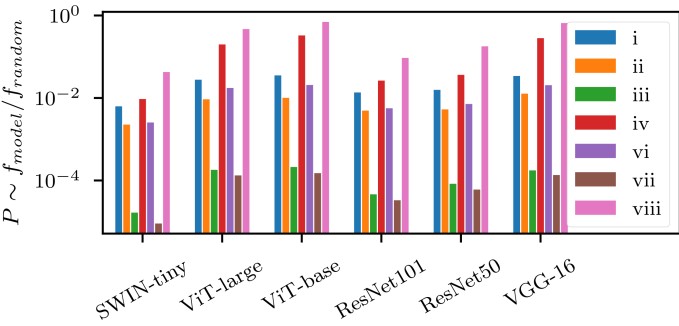

(c) Adjusted likelihood of violating each rule.

Figure 3: Errors arising due to counting or localizing.

likely to be violated. This points to the fact that the ability of models to adhere to domain constraints is somewhat dependent on the nature of the constraints themselves.

To summarize, we have demonstrated that although standard deep learning approaches appear to perform well if gauged by standard metrics on the VALUE Dataset (see Table 1), they fall short when it comes to domain constraint adherence. Even in the best performing models, 5.2% of predictions (up to 30% in the worst case) contain logical inconsistencies which would be unacceptable in critical applications and leads to the lowered effective F1 score (sF1). Furthermore, we showed that standard deep learning approaches demonstrate limited capacity to learn constraint information from the data itself, but that is largely dependent on the type of constraint. Further investigation is needed to explore this relationship between the types of constraints and ease of adherence. It is also unclear whether the improved performance of some models (Table 1) is owing to their ability to learn representations that are aligned with the constraints from the dataset or are meretricious. Additionally, it would be compelling to explore if we can evaluate the generated counterfactual in a deep SCM framework (Pawlowski et al., 2020; Saha and Garain, 2022) using the rules presented in the paper. In other words, while the axiomatic soundness of counterfactual image models were tested (Monteiro et al., 2023), there is scope to test the logical soundness of these models. Model robustness also warrants further inspection especially since incorporating domain-knowledge has shown initial promise in improving adversarial robustness (Melacci et al., 2022).

## 6 Conclusions

Domain knowledge in the form of logical or numerical constraints arise in many critical applications, and it is not clear how these rules are to be incorporated into deep learning systems. Owing to the great practical relevance, this area has been widely studied, but the lack of high quality datasets featuring a diverse curated set of rules has hindered thorough analyses and development of techniques. To address this we introduced the VALUE Dataset, which consists of 200,000+ images of chess games in progress, with the task of recreating the state of the chess board. Additionally, we curated a set of simple and computationally efficient sanity-check rules to ensure that the predictions are consistent with the rules of chess. In addition to standard performance metrics, suitable metrics were devised to analyze performance of deep learning systems in regard to domain constraint obedience. We explored the performance of several popular and state-of-the art vision models to show that although they appear reasonably accurate, they produce a plethora of inconsistencies, indicating that these systems lack the ability to learn underlying semantic structure from data alone. We further utilized our rule set to highlight key abilities and weaknesses of current approaches in deep learning.

Further research is needed to develop a standard framework for incorporating domain constraints into deep learning systems. We have made all necessary materials publicly available to aid future work on developing techniques to address this pressing challenge.

## Broader Impact Statement

Our work is intended to enhance exploration of the ability of deep learning based systems to adhere to domain constraints, which could lead to making these systems more deployable in key sectors like healthcare, robotics, etc. Thus our work, which tries to improve the understanding of the shortcomings of deep learning based systems, can potentially lead to these systems being deployed with more reliability and trust to several automated pipelines down the road. Thus, this has the same perils and advantages that enhanced automation brings to society. On one hand, it improves productivity and access and reduces costs. On the other hand, it helps in automating certain tasks that could render some jobs obsolete which has adverse consequences in societies without strong social safety nets.

## Acknowledgments and Disclosure of Funding

This research is partially supported by the Indo-French Centre for the Promotion of Advanced Research (IFCPAR/CEFIPRA) through Project no. 6702-2. Additionally, the authors would like to thank Ms. Sutanoya Chakraborty for her help in reviewing and editing the manuscript.

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
