# OpenReview forum: "VALUED - Vision and Logical Understanding Evaluation Dataset"
_DMLR — Accepted by DMLR_

### Review · Reviewer_jbGi · 2024-06-02

**Recommendation:** 2
**Confidence:** 2

**Summary Of Contributions:**

The paper introduces the VALUE (Vision And Logical Understanding Evaluation) Dataset, consisting of over 200,000 annotated images of chess games. The dataset aims to evaluate and improve the logical coherence of vision models by incorporating a set of logical constraints based on the rules of chess.

**Strengths:**

Please see above.

**Audience:**

Yes

**Broader Impact Concerns:**

- Enhancing the logical reasoning of vision models could lead to job displacement in industries without strong social safety nets.
- The ethical implications of deploying such systems in critical areas like healthcare and robotics need more thorough consideration.

**Claims And Evidence:**

The authors claim that existing vision models struggle with logical consistency and that the VALUE dataset provides a means to evaluate and improve this aspect. This is supported by the analysis of model performance, revealing logical inconsistencies in predictions.

**Datasets And Benchmarks:**

The VALUE dataset provides a large, annotated dataset with logical constraints derived from chess. However, the focus on chess limits its broader applicability.

**Extended Submissions:**

NA

**Limitations:**

- The dataset is specifically tailored to chess, limiting its applicability to other fields.
- The logical constraints set is not exhaustive, potentially overlooking other significant logical rules.

**Requested Changes:**

- Provide a more detailed explanation of the image rendering process and the introduction of noise.
- Include a broader set of logical constraints to enhance the dataset's robustness.
- Perform more extensive fine-tuning of baseline models.
- May discuss the potential generalizability of the findings to other domains beyond chess.

**Strengths And Weaknesses:**

Strengths:

- The dataset is extensive, with over 200,000 annotated images.
- The inclusion of logical constraints as an evaluation metric is a novel approach.
- The authors provide a detailed analysis of several state-of-the-art vision models, identifying their logical reasoning capabilities.



Weaknesses:

- The focus on chess may limit the generalizability of the findings to other domains.
- Methodological details, such as image rendering and noise introduction, are not thoroughly explained.
- Baseline results are primarily on pre-trained models without extensive fine-tuning, which may not fully demonstrate the potential improvements achievable with the dataset.
- The set of logical constraints is not exhaustive and may miss other critical logical rules.
- The broader impact and ethical implications of deploying such systems are not sufficiently addressed.

---

### Review · Reviewer_uBju · 2024-06-07

**Recommendation:** 3
**Confidence:** 3

**Summary Of Contributions:**

The paper proposes the VALUE (Vision And Logical Understanding Evaluation) dataset that consists of real-world chess images together with their rule abstractions. The dataset is used for evaluating vision models' reasoning capabilities for localization and enumeration. The experimental results show that existing models fail to perform well on the reasoning tasks.

**Strengths:**

The strengths have been described in the "Strengths And Weaknesses" section.

**Audience:**

Yes

**Broader Impact Concerns:**

The dataset does not have many concerns about negative societal impacts.

**Claims And Evidence:**

The paper needs to provide more pieces of evidence as stated in the "Strengths And Weaknesses" section to validate their benchmark.

**Datasets And Benchmarks:**

Yes, the details are well described. However, authors need to address the concerns listed in the sections above. Given the comprehensiveness of the authors' response, I change my rating from 2 (Lean to Reject) to 3 (Lean to Accept).

**Extended Submissions:**

N.A.

**Limitations:**

The limitations have been described in the "Strengths And Weaknesses" section.

**Requested Changes:**

The requested changes have been described in the "Strengths And Weaknesses" section.

**Strengths And Weaknesses:**

**Strength**
1. The paper is well-written with a clear storyline that motivates the work.
2. The research problem of model reasoning capabilities is important.
3. The dataset is large-scale and diverse.

**Weakness**
1. The motivation of the dataset potentially overlaps with many existing benchmarks for Neuro-Symbolic AI and reinforcement learning. Several existing datasets such as Visual Sudoku / Atari Games / HighWay can also achieve the same evaluation objective as the VALUE dataset. The authors need to argue sufficiently the uniqueness of their benchmark compared to existing Neuro-Symbolic ones.

2. The paper should consider evaluating foundational vision-language models (e.g., BLIP-2, LLaVA) on this benchmark.

3. More visualizations and demonstrations of the dataset are needed in the paper to validate the data quality.

---

### Review · Reviewer_KEij · 2024-06-28

**Recommendation:** 3
**Confidence:** 2

**Summary Of Contributions:**

This paper proposes a task and dataset, VALUED, which evaluates the ability of vision models to make predictions that abide by a rich set of domain-specific constraints. In particular, VALUED consists of images of chessboards produced from real chess games, and the goal is for a vision model to output the true chessboard state and layout from the image. Framed as a classification problem, even if the model generally can classify what piece is on each square, it often produces a set of predictions that collectively violate several constraints of valid chessboard states (for instance, there should be one king, and the two kings should not be on adjacent squares)---despite the fact that all images in the training dataset consist of valid chess games. The paper proposes several metrics that allow for finer-grained analysis of whether constraints are followed, and fine-tune several vision models to demonstrate that while their raw F1 scores appear high, only considering the predictions that are "sane" chessboard states significantly lowers these scores. Analysis is also provided on what sorts of constraints are easier to follow (localizing versus counting). VALUED is the first dataset of its kind in terms of dataset size and diversity/coverage of rules.

**Strengths:**

See above.

**Audience:**

Yes

**Broader Impact Concerns:**

Broader impacts are discussed, and I do not have any concerns about potential negative impact.

**Claims And Evidence:**

Yes

**Datasets And Benchmarks:**

Details for collection, and availability/maintenance are provided. URL is provided as well as documentation.

**Extended Submissions:**

N/A

**Limitations:**

See above for weaknesses.

**Requested Changes:**

* In section 3.1, there is a mention of more rules ("b can't be trapped in the last rank behind 3 adjacent p"). It would be helpful to have a few examples of these as future work or listed along with the dataset as desired features so that people can contribute.
* More discussion about how to incorporate domain constraints for VALUED beyond the vanilla fine-tuning baseline.
* Provide explanation of figure 2 and if different properties of the image can be modified; if so, what impact does it have on overall versus sane metrics?

**Strengths And Weaknesses:**

Strengths:
* Large, high-quality dataset with generalizable procedure to create even more samples with varying properties.
* Proposes a diverse rule set that covers a sufficient part of the domain.

Weaknesses:
* Could benefit from more discussion about what more complex rules could be incorporated into the rule set
* Could benefit from more discussion about how to incorporate VALUED's domain knowledge constraints into training the model. The related work briefly mentions there are ways to transform the input data, the loss function, and the training procedure to incorporate constraints, but it would be interesting to discuss as future work---or even implement---some simple baselines in the context of the chessboard state problem.
* It is unclear how the construction of images plays a role in the model's performance. In figure 2, there is some mention of occlusion and object density. Are these parameters that can be set in the dataset construction process? More generally, I am curious if making the images more challenging (e.g., more occlusion, or changing the rotation of the board) would uniformly degrade performance or impact the sane metrics more.
* An interpretation of the results here is that ~200K+ samples is not enough for these vision models to learn a rule like "no pawn on the first or last rank." I'm curious how VALUED could be extended to smaller chess variants (https://en.wikipedia.org/wiki/Minichess) or simpler state-based board games where the output space is a bit smaller. We then could maybe be able to observe performance scaling in the complexity of the board and the size of the rule set. This is largely beyond the scope of this paper though.

---

### Review · Reviewer_xGc5 · 2024-07-20

**Recommendation:** 3
**Confidence:** 2

**Summary Of Contributions:**

The paper introduces the VALUE (Vision And Logical Understanding Evaluation) Dataset, a collection of over 200,000 annotated images of chess games. The dataset aims to address the gap in evaluating logical understanding in deep learning-based computer vision systems by incorporating a curated set of logical constraints. The primary contributions include:

1. Creation of a large-scale dataset with annotated images of chess games.
2. Development of a curated rule set to enforce logical constraints on predictions.
3. Introduction of additional performance metrics to measure logical consistency.
4. Analysis of the performance of popular vision models on this dataset, highlighting their limitations in logical understanding.

**Strengths:**

See above

**Audience:**

Yes

**Claims And Evidence:**

Yes

**Datasets And Benchmarks:**

YES

**Extended Submissions:**

N/A

**Requested Changes:**

See Weakness

**Strengths And Weaknesses:**

S1 Innovative Dataset: The introduction of a dataset focused on logical understanding in vision tasks is a significant and innovative contribution.

S2 Scale and Quality: The large size and detailed annotations of the dataset make it a valuable resource for researchers.

S3 Logical Rules and Metrics: The curated rule set and new metrics provide a robust framework for evaluating the logical consistency of models.

S4 Comprehensive Analysis: The baseline performance analysis of popular models offers valuable insights into their strengths and weaknesses concerning logical reasoning.



W1 May incorporation of More Complex Rules: The rule set, while effective, could be expanded to include more complex constraints such as tactical motifs or strategic patterns in chess. This would provide a more comprehensive evaluation of logical reasoning capabilities.
Training

W2 Methodologies: The paper briefly mentions methods for incorporating domain knowledge into the training process but does not provide detailed methodologies or examples. Future work could explore transforming input data, modifying loss functions, or adjusting the training procedure to incorporate these constraints.

W3 Occlusion and Object Density: The impact of image construction, such as varying levels of occlusion and object density, on model performance is not deeply analyzed. This is crucial as real-world scenarios often involve such complexities.

W4 Scalability to Other Domains: While the dataset is comprehensive for chess, its principles should be tested on simpler board games or different domains to understand performance scaling with complexity and rule set size.